# Systematic Characterisation of the Fragmentation of Flavonoids Using High-Resolution Accurate Mass Electrospray Tandem Mass Spectrometry

**DOI:** 10.3390/molecules29225246

**Published:** 2024-11-06

**Authors:** Candy Jiang, Paul J. Gates

**Affiliations:** School of Chemistry, University of Bristol, Cantock’s Close, Bristol BS8 1TS, UK

**Keywords:** flavonoids, mass spectrometry, fragmentation, structure characterisation

## Abstract

Flavonoids are a class of polyphenolic secondary metabolites found in plants. Due to their ubiquity in our daily dietary intake and their major anti-oxidative, anti-inflammatory and anti-mutagenic activities, they have been a major focus of wide-ranging research for the past two decades. Mass spectrometry combined with liquid chromatography is one of the most popular techniques for the analysis of flavonoids. In this study, high-resolution accurate mass electrospray tandem mass spectrometry was used to study 30 flavonoids in both positive and negative ionisation modes. From the data obtained, common losses were summarised and compiled. Dominating neutral losses were tabulated. The radical loss of CH_3_· was observed in flavonoids containing methoxy groups and three key diagnostic product ions were identified. These were *m*/*z* 153 (indicative of two OH groups on ring A) *m*/*z* 167 (indicative of one OH and one methoxy group on ring A) and *m*/*z* 151 (a flavanol, with no ketone oxygen but two OH groups on ring A). These will be useful in structural elucidation of unknown flavonoids and flavonoid metabolites. Energy breakdown graphs were utilised to distinguish between three pairs of structural isomers, and to help rationalise proposed fragmentation pathways. Lastly, a competition of loss of CH_3_· and methane was reported for rhamnetin and isorhamnetin in the negative ion mode for the first time. Proposed fragmentation pathways were given to rationalise the differences in peak intensities for this competitive process.

## 1. Introduction

Flavonoids are a class of polyphenolic secondary metabolites mainly found in plants [1]. They are present in fruits, nuts and vegetables and have become a part of our daily dietary intake. Flavonoids often contribute to the colour of flowers, where they assist in pollination and provide UV protection [2]. Plants use them to aid growth and as a defence against disease and infection [2] and, due to their unique structures, flavonoids have been found to have many important biological activities such as anti-oxidative, anti-inflammatory, anti-mutagenic and anti-carcinogenic [3,4,5].

For example, the isoflavonoid glabridin from *Glycyrrhiza glabra* has been found to inhibit low-density lipoprotein oxidation by scavenging free radicals [6]. The human metabolism of flavonoids has been studied and reviewed in detail [7], along with the mechanism for their anti-oxidant activity, absorption and bioavailability [8,9,10]. It has been noted that the configuration and number of hydroxyl groups and the substitution of the various functional groups on the structures of individual flavonoids affect their bioavailability, metabolism and biological activities [11]. Due to their diverse range of chemical and biological properties, their extensive applications include nutraceutical, pharmaceutical, medical and cosmetic uses [12,13].

Increasing awareness of the health benefits and pharmaceutical applications of flavonoids has led to a considerable increase in the number of studies of flavonoids using mass spectrometry (MS). The amount of the available literature on flavonoid analysis by various MS techniques can be overwhelming. The earlier studies of flavonoids were performed using electron ionisation (EI), methane chemical ionisation (CI) and fast atom bombardment (FAB) MS [14,15,16,17]. Results stated that the outcome of the spectra was heavily dependent on the selection of the MS technique. Peak intensities of specific ions in the resulting mass spectra depended on the ionisation technique and the structure of the analytes.

As the application of electrospray ionisation (ESI) became mainstream, studies started to implement ESI coupled with high-performance liquid chromatography–tandem mass spectrometry (HPLC-MS/MS) to investigate various classes of flavonoids and their substitutes [18,19]. Recent advances in the analytical approach for the study of flavonoids using ESI and other modern MS techniques have been readily and widely reviewed [20,21,22]. Other analytical methods looked into coupling unified chromatography and ultra-performance liquid chromatography with ESI to look into various flavonoid groups [23,24]. Direct analysis in real time (DART) coupled with MS is also employed to look into flavonoid content in bee products for food quality purposes [25].

From the MS/MS studies of flavonoids, specific fragmentations such as dehydration and loss of CO from protonated molecular ions were predominantly found [17]. The fragmentation of flavonoids was explored in the negative ion mode [26,27]. Several fragmentation pathways were proposed based on highly specific negative product ions due to the retro Diels–Alder (RDA) process. Results from these studies demonstrated that, in the analysis of flavonoids, the positive ion mode gives more structural information, but, when combined with the negative ion mode, could provide sufficient information for identifying unknown flavonoids compounds and new metabolites without resorting to challenging purification methods [28,29].

One common challenge in the analysis of flavonoids is the vast number of variations in their functional group’s position, mainly in the locations of the hydroxy and the methoxy groups on the side rings. This results in a dominating presence of structural isomers within the same flavonoid subclasses. Since all MS techniques are based on mass-to-charge ratio (*m*/*z*) separation, separating these structural isomers is challenging. Errors resulting from unreliable peak separation or spectra interpretation are common. Therefore, after examining the existing literature, this study selected 30 commercially available flavonoids from 6 flavonoid classes and performed a systematic MS/MS analysis in both the positive and negative ion modes using ultra-high-resolution accurate mass ESI-MS/MS.

This systematic approach enabled the identification of three key diagnostic product ions, which can be used as fingerprints to investigate unknown flavonoids. Integration of the positive and negative ion modes provided complementary data. Energy breakdown graphs were sketched to demonstrate the relationship between collision energy and peak intensity of the product ions of selected isomeric flavonoid pairs, and to display distinctive features for each flavonoid that would otherwise be overlooked. Complete fragmentation pathways were also proposed for selected flavonoids. This paper also investigated the competition between neutral loss and radical loss observed for rhamnetin and isorhamnetin due to the position variation of the methoxy moiety.

## 2. Results and Discussion

Based on the previous literature, [30] a numbering system is adapted in this paper. Figure 1 shows the basic structure of a flavonoid. Ring A is marked in red, ring B in blue and ring C in black. The atom numbering starts from the heterocyclic oxygen in ring C and ends on the ring B benzene carbon. A nomenclature is also added to clarify the assignment of the product ions. For example, a ring cleavage labelled ‘**rC^x,y^_AorB_**’ represents the cleavage location on ring C between carbon–carbon positions x and y, leaving either ring A or B intact.

### 2.1. Common Losses in the Positive Ion Mode

Positive ion mode ESI-MS/MS spectra are collected for all 30 flavonoids, and the neutral/radical losses from the precursor ions are tabulated (see Table A2 in Appendix A). The first trend to note is that flavonoids from the same group do not always have the same common losses, although there are degrees of similarity. Secondly, common losses are mostly dependent on the location and number of –OH groups (loss of H_2_O), –MeO groups (loss of CH_3_ radical) and the presence of a C=O function in ring C (loss of CO, mass 28). Any losses higher than 100 mass units result from ring cleavage through the RDA processes on ring C. Table 1 summarises the typical neutral losses observed in the positive ion mode.

From the common losses, three key diagnostic product ions are found to be directly related to the structure of their precursor ion (PI). These are *m*/*z* 167, *m*/*z* 153 and *m*/*z* 151. Among these, *m*/*z* 153 is the most commonly observed and results from cleavage on ring C between atoms 1 and 4, leaving ring A intact, and is therefore designated as rC^1,4^_A_. Product ion *m*/*z* 167 is also due to cleavage on ring C between atoms 1 and 4, leaving ring A intact, and is therefore designated as rC^1,4^_A_. The difference between these two ions is due to a methoxy group on ring A. Lastly, ion *m*/*z* 151 is formed from cleavage on ring C between carbons 3 and 10, therefore designated as rC^3,10^_A_. All three product ions are listed in Table 2, along with the flavonoids (and their classes) that produced them.

Table 3 is a demonstration of the three key diagnostic properties of the product ions allowing for the observation of the structural correlation between precursor and product ions. Quercetin, sakuranetin and epigallocatechin are used as examples. All three product ions are coloured to highlight the exact structural resemblance to their respective PIs. Therefore, it can be concluded that the presence of *m*/*z* 153 demonstrates the presence of two OH groups in ring A (at C-6 and 8) and a C=O at C-4 in ring C. Observation of *m*/*z* 167 indicates the presence of an OH group at C-6, a methoxy group at C-8 and a ketone at C-4 on ring C. Finally, product ion *m*/*z* 151 shows the presence of no ketone on ring C i.e., C-4 is an unoxidised CH_2_, but two OH groups on ring A at C-6 and C-8. The presence of these product ions gives a clear insight into the structures of their respective flavonoid precursor ions. These findings will be useful in the structural elucidation of unknown flavonoids.

### 2.2. Energy Breakdown Graphs for Isomeric Differentiation

Energy breakdown graphs (EBGs) are plots of collision energy versus relative peak intensity for the selected product ions. They can demonstrate the different energy thresholds for alternative fragmentation routes and showcase secondary losses during the fragmentation processes. Because of this, they have been used previously for a variety of MS^n^ studies as a complementary analytical tool to provide additional information of the fragmentation processes [31,32,33]. EBGs have previously been used in the study of flavonoids to examine how changes in collision energy affect peak intensities for selected product ions [34]. Due to the unique ability of EBGs to display a complete picture of the relationship between product ion peak intensities vs. collision energy, they have also been employed for isomer differentiation [35,36,37,38]. Product ions of a pair of structural isomers can often exhibit different peak intensities under the changes in collision energy. The same pair of structural isomers could also generate unique product ions as the collision energy increases. These specific characteristics can all be captured and visualised by EBGs. Therefore, EBGs are a powerful tool in addition to MS/MS spectra. In this part of the paper, EBGs are plotted for three pairs of isomeric flavonoids from their MS/MS spectra in the positive ion mode.

#### 2.2.1. Rhamnetin and Isorhamnetin

Rhamnetin and isorhamnetin are structural isomers differentiated by the location of the methoxy group (see Figure 2 for structures). Each of them has one methoxy group and four OH groups. Their energy breakdown graphs shown in Figure 2 indicates a few differences. Firstly, there are fewer product ions for isorhamnetin. Secondly, the difference in the intensity of product ion *m*/*z* 302 is almost four times more intense for isorhamnetin compared with rhamnetin. These observations are both due to the intensity of the loss of the methyl radical. In the case of rhamnetin, the losses of H_2_O and CO compete and are a much preferred fragmentation route. Except for the common losses of the methyl radical and of a CO, rhamnetin and isorhamnetin have noticeably different fragmentation pathways despite being structural isomers.

The fragmentation pathways and proposed structures of the product ions observed in the analyses of rhamnetin and isorhamnetin are shown in Figure 1 and Figure 2. The protonation site is on the carbonyl oxygen of ring C unless stated otherwise. Figure 1 shows multiple losses of CO from rhamnetin, either after an initial loss of H_2_O or directly from the precursor ion. The competition between the loss of a methyl radical and other routes makes this route less favourable. The two ring cleavages are marked in purple. The first of which occurs after a water loss to produce *m*/*z* 179, which is the third most intense peak in the EBG. The second ring cleavage results in *m*/*z* 167. However, this peak can also be produced through a loss of CO along with a C–O bond cleavage. Both routes go on to further lose H_2_CO to produce *m*/*z* 137, with the structure at the bottom left of the scheme being proposed as being the most likely.

Isorhamnetin has fewer product ions than rhamnetin, with *m*/*z* 302 as the most prominent. This can be explained by the formation of a stabilised radical, the structure of which is shown in Figure 2. Compared with the structure of ion *m*/*z* 302 for rhamnetin, *m*/*z* 302 for isorhamnetin is stabilised by the adjacent hydroxy group and conjugation through ring B and C. The production of *m*/*z* 274 further proves this, as product ion *m*/*z* 302 is stable enough to undergo secondary fragmentation by loss of CO. A ring closure is proposed for ion *m*/*z* 274 to explain why it does not fragment further. The second most intense peak is *m*/*z* 285, which is the result of the loss of methanol. Protonation is proposed to be on the methoxy oxygen to induce an ortho elimination. This specific loss for isorhamnetin is indicative of the presence of a methoxy group on ring B at the ortho position, agreeing with the previous literature [34]. Kaempferide, which has a methoxy group at the same location, exhibits the exact same mechanism [39].

The structural differences between rhamnetin and isorhamnetin also affect the location of ring cleavages. Isorhamnetin has two ring cleavages: _r_C^4,10^_A_ and _r_C^1,4^_A_. This is very different to rhamnetin, and it shows that, even though both flavonols have the same number of OH groups and a methoxy group, the location of these functional groups significantly affects the formation of the product ions. The relative stability of the different radical ions produced influences the entire spectra. The stability of the ring B product ions also influences which fragmentation pathways would be followed.

#### 2.2.2. Kaempferol and Fisetin

As with rhamnetin and isorhamnetin, both kaempferol and fisetin belong to the flavonol class. Both kaempferol and fisetin have four OH groups in their structures. However, kaempferol only has one OH group on ring A, whereas fisetin has two. The converse is true for ring B (see structures in Figure 3). Differences in their product ions can be viewed in their energy breakdown graphs. Kaempferol has a dominant ion at *m*/*z* 177, resulting from a ring cleavage, whereas fisetin favours the loss of mass 46 to produce the ion at *m*/*z* 241. Product ion *m*/*z* 177 occurs at a much lower collision energy than the others for kaempferol. All product ions of fisetin occur at a collision energy of around 23 eV, as seen in the energy breakdown graph.

The fragmentation pathways and proposed structures of the product ions observed in the analyses of kaempferol and fisetin are shown in Figure 3 and Figure 4. The following discussion aims to rationalise the proposed product ion structures and their EBGs. Kaempferol has extensive ring cleavages, and the bonds broken are coloured in purple. The high peak intensity of product ion *m*/*z* 177 could be due to a stabilised five-membered ring structure with a tertiary carbocation. The key diagnostic product ion *m*/*z* 153 is present with kaempferol. Extensive H_2_O and CO losses are also observed.

Fisetin follows the same number of H_2_O and CO losses to produce product ions *m*/*z* 269, 259, 241, 231 and 213. From its EBG, the ion at *m*/*z* 241 is the most intense and occurs at much lower energy than the others. This suggests that it may be due to the loss of formic acid (HCOOH) in one step, as highlighted in Figure 4, in addition to the first loss of H_2_O, followed by a CO. It then goes on to lose a CO to produce the ion at *m*/*z* 213. This agrees with the observation that the ion *m*/*z* 213 is the second most intense peak in the EBG. Compared with kaempferol, it has one less ring cleavage, and both product ions from these cleavages leave ring B intact. Product ion *m*/*z* 149 has two fragmentation pathways, with one occurring directly from the cleavage of ring C at atoms 1 and 4, the other is by a carbon–carbon bond breakage from ion *m*/*z* 259. The first is more likely according to the EBG. Both routes can be followed by another loss of CO to give product ion *m*/*z* 121. Finally, the ion at *m*/*z* 185 has the lowest intensity and requires the highest collision energy. This can be explained by a less favourable charge remote fragmentation and the need to break the already stabilised ring structure of *m*/*z* 213.

Kaempferol and fisetin differ in their locations, numbers and remaining structures of the ring C cleavages. Product ions of these ring cleavages are key to their structural identification. Kaempferol has a key diagnostic product ion *m*/*z* 153, which can distinguish it from fisetin due to the location of the OH group.

#### 2.2.3. Chrysin and Daidzein

The final pair of isomeric flavonoids are chrysin and daidzein. They are selected as they are structural isomers but belong to different flavonoid classes. Both of their protonated molecular ions occur at *m*/*z* of 255. Their highly distinctive EBGs are presented in Figure 4. The most intense peak for chrysin is the product ion *m*/*z* 153, the key diagnostic peak from a specific structure on ring A. This peak also first appears at a lower collision energy compared with others. In contrast, daidzein has three product ions at the lower collision energy. These are *m*/*z* 199, 137 and 227.

The proposed fragmentation scheme for chrysin and daidzein are shown in Figure 5 and Figure 6. Chrysin has two ring cleavages, including the key diagnostic product ion for ring structure A. Also, it has a loss of formic acid to produce ion at *m*/*z* 209. There are also extensive ring contractions for chrysin, which agrees with the previous literature [40]. The product ion *m*/*z* 187 is created from a very unusual loss of carbon suboxide (C_3_O_2_,), first observed in 2001 for flavones and later in a few other studies both in positive and negative ion mode [40,41,42]. Although no mechanisms have been proposed. This could be an opportunity for future work to investigate a viable mechanism. Also, a loss of CH_2_CO gives three possible structures for product ion *m*/*z* 213. Structure **3** has been shown previously in the literature [40]. However, we believe structure **1** is more stabilised.

Daidzein has only one ring cleavage from the protonated PI, which results in product ion *m*/*z* 137. The most intense peak *m*/*z* 199 is produced from the loss of a CO from *m*/*z* 227. Two possible structures are proposed for this ion; both involve a ring contraction to form a cyclopentadiene in their structure, depending on which CO is lost. Product ion *m*/*z* 145 has the lowest intensity, possibly due to a less favourable charge remote ring cleavage from ion *m*/*z* 237. In summary, for both chrysin and daidzein, ring cleavage on ring C is very energetically favourable. Chrysin has extensive ring contractions that result in the unusual neutral loss of C_3_O_2_. Daidzein can also undergo ring contraction to lose a CO.

### 2.3. Common Losses in the Negative Ion Mode

MS/MS data for the 30 flavonoids in negative ion mode are also investigated, and neutral/radical losses of their deprotonated precursor ions are summarised in Appendix A
Table A3. Flavonoids are grouped and marked by different colours based on their classes, as in the positive ion mode. The first trend is consistent with the observations in the positive ion mode. Flavonoids from the same classes do not always exhibit the same common losses. They only have the same common losses if they are structural isomers from the same flavonoid class. For example, in the flavone group, genkwanin and diosmetin both only produce a loss of 15, which corresponds to a CH_3_· radical loss. Gallocatechin and epigallocatechin, and catechin and epicatechin, are two pairs of structural isomers that are all flavanols; hence, they have the same neutral losses.

Secondly, compared with the positive ion mode CID MS/MS data, there are more lower mass neutral losses. The most common ones are summarised in Table 4. Like with the positive ion mode, common losses in the negative ion mode are also dependent on the hydroxy and methoxy group locations. Other than the predominantly occurring loss of H_2_O, loss of CO is frequently observed. Any losses with higher than 100 mass units is likely to be an indication of ring cleavage on the ring C.

Results from the negative ion mode MS/MS have one major difference from the positive ion mode spectra: the loss of 16, which corresponds to a CH_4_ methane loss. This has been previously published in the literature of this research group [43], and the data acquired here are strong evidence to further support this observation. Flavonols with a neutral loss of CH_4_ are highlighted in the box in Table A3. These are discussed and investigated further using energy breakdown graphs and proposed mechanisms.

### 2.4. Loss of CH_3_· and CH_4_

Both rhamnetin and isorhamnetin have a loss of 16, corresponding to the loss of methane confirmed by accurate mass measurement (see Table A10 and Table A11 in Appendix B). This observation of CH_4_ elimination is very interesting and has been published before with heterocyclic aromatic amines [44] and the flavanone hesperetin but not with any other flavonoids. Hence, it remains the focus of the last part of this paper.

Out of all the flavonoids with methoxy groups, only rhamnetin and isorhamnetin exhibit a loss of CH_4_ in their negative ion MS/MS spectra. However, all the flavonoids mentioned share a common loss of a CH_3_· methyl radical. This could be because the unique CH_4_ loss requires a hydroxyl group on the meta carbon in ring C, which the flavone groups lack. An OH group on the neighbouring carbon is also essential to facilitate this mechanism. This explains why, although kaempferide has a hydroxyl group on the meta carbon in ring C and a methoxy group on ring B, the lack of an OH group on the same ring prevents the loss of 16.

Figure 5 shows the negative ion MS/MS spectra for both rhamnetin and isorhamnetin. Although rhamnetin and isorhamnetin are structural isomers and belong to the same flavonol group, their negative ion MS/MS spectra are distinctively different. They both have losses of CH_3_· (*m*/*z* 300) and CH_4_ (*m*/*z* 299). Compared with isorhamnetin, *m*/*z* 299 of rhamnetin is at a much higher intensity and is slightly more intense than *m*/*z* 300. In contrast, for isorhamnetin, the loss of CH_3_· is a much more favourable route. For rhamnetin, a number of additional fragment routes are followed to generate losses of CO in much the same way as in positive ion mode. For isorhamnetin, the only additional product ion is *m*/*z* 285 (loss of MeOH from the PI), occurring at a very low intensity. The loss of the methyl radical totally dominates this spectrum.

Figure 6 shows distinctive differences in the product ions for these two flavonols despite the similarity in their structures. Rhamnetin produces more intense product ion peaks, especially in the lower mass range (*m*/*z* 165). Whereas the loss of CH_3_· dominates the isorhamnetin spectrum. In the breakdown graph for rhamnetin, peak *m*/*z* 299 starts to occur at CID energy 15eV and *m*/*z* 300 at just above 20 eV. As the collision energy increases, *m*/*z* 299 peak intensity also increases. This supports our theory that loss of CH_4_, in the case of rhamnetin, is more energetically favourable than the loss of CH_3_. Figure 6 shows very low intensity for the loss of CH_4_ for isorhamnetin and other product ions, except *m*/*z* 300. This observation could be supported by a different mechanism for the loss of CH_4_, which is discussed next. The striking differences in the mass spectra between these two isomeric flavonols in the negative ion mode build an excellent foundation for isomerisation differentiation. The unique loss of CH_4_ could be used for the structural elucidation of unknown flavonoids in future work.

Figure 7 and Figure 8 are the proposed fragmentation pathways for rhamnetin and isorhamnetin, including the proposed alternative mechanisms for the generation of product ions *m*/*z* 300 and 299 in the negative ion mode. In Figure 7, the negative charge is proposed to be on the meta oxygen in ring C, facilitating a 1,5 hydride shift to eliminate CH_4_ on ring A. This also promotes a ring opening between C3 and C4 on ring C, resulting in the structure for product ion *m*/*z* 299 in Figure 7. Fragmentation of *m*/*z* 299 further produces *m*/*z* 271, 193 and 165. For isorhamnetin, due to the different positions of the MeO and OH group, an alternative mechanism is proposed, as published in the literature for hesperetin [44]. Unlike the 1,5 hydride shift, this mechanism is less energetically favourable and could be a reason for the lower intensity of peak *m*/*z* 299 for isorhamnetin. The charge remote fragmentation and the lack of ring opening could also contribute to this lower intensity in Figure 5.

## 3. Materials and Methods

In this study [45], 30 flavonoids from 6 subclasses were analysed. They were all obtained from Sigma Aldrich (purity ≥ 90%). The flavonoids were dissolved in water/methanol (1:1, 1 mg/mL) to make up standard solutions. The flavonoids were quercetin, morin, rhamnetin, isorhamnetin, kaempferide, diosmetin, kaempferol, fisetin, 5-hydroxyflavone, galangin, baicalein, apigenin, luteolin, scutellarein, chrysin, daidzein, 6-hydroxyflavanone, 7-hydroxyflavonone, gallocatechin, epigallocatechin, catechin, epicatechin, aromadendrin, eriodictyol, genistein, sakuranetin, naringenin, genkwanin, hispidulin and wogonin.

Positive and negative ion ESI-MS/MS analyses were performed on an Orbitrap Elite mass spectrometer (Thermo Fisher Scientific, Hemel Hempstead, UK) using a heated ESI source. The analyte solutions were diluted to 0.1 mg/mL in methanol/water (1:1) prior to analysis and were delivered using the autosampler module of an RS3000 UHPLC system (Dionex, Hemel Hempstead, UK) at a flow rate of 10 μL/min. Tandem mass spectra were recorded in CID-MS/MS mode on isolated precursor ions (2 *m*/*z* window) with collision energies ramped from 0 to 50 eV. Energies were increased by 1 eV every 12 s. The resulting runs were 10 min per sample. A full scan (50–500 *m*/*z*) was obtained then the selected precursor ion was automatically selected in data dependent mode, isolated and underwent CID-MS/MS using dry N_2_ as the collision gas. For all experiments, the acquisition time was 0.25 min per scan using FTMS mode at a resolution of 240,000. Two microscans were summed with a maximum ion accumulation time of 200 ms.

## 4. Conclusions

This study systematically investigates the fragmentation of 30 flavonoids using ESI (positive and negative ion mode) high-resolution accurate mass MS/MS. Common losses are summarised and compiled. Neutral losses such as H_2_O and CO are dominating. Radical loss of CH_3_· is consistently observed for flavonoids having methoxy groups in their structures in both positive and negative ion modes. In addition, in the positive ion mode, three key diagnostic product ions are identified: *m*/*z* 153, indicative of two OH groups on ring A; *m*/*z* 167, indicative of one OH and one methoxy group on ring A; and *m*/*z* 151, a flavanol, with no ketone oxygen but two OH groups on ring A. These findings could provide essential insight into the structural elucidation of unknown flavonoids and flavonoid metabolites.

The addition of energy breakdown graphs to the conventional tandem mass spectra dataset has proven to be a powerful analytical tool in isomer differentiation in the case of flavonoid studies. Three pairs of structural isomers are selected for detailed discussion using their MS/MS spectra and energy breakdown graphs to help visualise the proposed fragmentation pathways. It is shown that, although they may share the same common losses, ring C cleavages are often structurally indicative due to their distinct A and B ring structures and substitution patterns. Isomeric flavonoids can display distinctive patterns in their energy breakdown graphs depending on functional group locations on ring A and B.

During the investigation of rhamnetin and isorhamnetin in the negative ion mode, both flavonols exhibit a highly unusual loss of methane apart from the usual radical loss of CH_3_·. This interesting finding initiates a more detailed examination on their MS/MS spectra and energy breakdown graphs. It is concluded that the loss of methane and radical loss of CH_3_· are in competition. By interpreting the proposed mechanisms for rhamnetin and isorhamnetin, the nature of this competition is revealed for the first time, and the observation of differences in peak intensities for these two routes could be rationalised.

The work carried out in this paper provides fundamental understanding and additional information to the existing flavonoid MS literature. The results presented will aid other flavonoid related research in many disciplines, including food quality control, pharmaceutical development and drug discovery. The combination of tandem mass spectrometry and energy breakdown graphs could also be applied to studies of other natural polyphenols, especially for the differentiation of isomers, structural elucidation and qualitative studies of these complex compounds.

## Data Availability

The raw data are available from the University of Bristol data repository, data.bris, at https://doi.org/10.5523/bris.1mt60eklqgii41zvvz1poadro6.

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
