# Peer review of "Systematic Characterisation of the Fragmentation of Flavonoids Using High-Resolution Accurate Mass Electrospray Tandem Mass Spectrometry"

_molecules, 2024, doi:10.3390/molecules29225246_

Round 1

Reviewer 1 Report

Comments and Suggestions for Authors

The manuscript Molecules-3269500 entitled “Systematic Characterisation of the Fragmentation of Flavonoids using High-Resolution Accurate Mass Electrospray Tandem Mass Spectrometry.” discusses the observed radical loss of CH3. in flavonoids containing methoxy groups and three diagnostic product ions which were identified for structural elucidation of unknown flavonoids and flavonoid metabolites from 3 different classes.

Comments-

It would be interesting to mention the three diagnostic product ions identified for structural elucidation of unknown flavonoids and flavonoid metabolites within the abstract.

The reference style mentioned in the text doesn’t match the one in the reference section. This made the understanding the introductory text from line 42 onwards difficult. I couldn’t relate to the references because I couldn’t find them. The text used numbered citation, while the reference section has the author names without numbers mentioned in the text.  

The “Error! Reference source not found.” Text was found at several places. This needs to be rectified for proper understanding.

 I was really interested to understand the references 21 and 22 for “Several fragmentation 58 pathways were proposed based on highly specific negative product ions due to the Retro 59 Diels-Alder (RDA) process. Results from these studies demonstrated that in the analysis 60 of flavonoids, positive ion mode gives more structural information, but when combined 61 with negative ion mode, it could provide sufficient information for identifying unknown 62 flavonoid compounds without resorting to challenging purification methods, and for 63 identifying new metabolites.”

For the text at - “One common challenge in the analysis of flavonoids is the vast number of variations 65 in their functional group's position, mainly in the locations of the hydroxy and the meth-66 oxy groups on the side rings. This results in a dominating presence of structural isomers 67 within the same flavonoid subclasses. Since all MS techniques are based on mass-to-68 charge ratio (m/z) separation, separating these structural isomers is challenging.” I would suggest to refer to a study for differentiation of isomers with methoxy substituents in ortho and para positions of the B-ring which was achieved using MS/MS in chalcones and flavanones. See - Rapid Commun. Mass Spectrom. 2013, 27, 1303–1310. Similarly, in related products like arylidene indanones this strategy also works well and gives the same results. - Rapid Commun. Mass Spectrom. 2013, 27, 2461–2471.

The fragmentation nomenclature generally used for flavonoids is well known and used by many (original refs 77 and 78 in Cuyckens, F.; Claeys, M. Mass Spectrometry in the Structural Analysis of Flavonoids. J. Mass Spectrom., 2004, 39, pp. 1-15. DOI: 615 10.1002/jms.622) and also in Rapid Commun. Mass Spectrom. 2013, 27, 1303–1310.  I could not trace the reference 23, which was identified in the manuscript due to the earlier problem I mentioned. *The reference style mentioned in the text doesn’t match the one in the reference section.

The suggested nomenclature is fine, as it is clearly demonstrated in the figures. But it would be good to understand why the authors decided to use this new one rather than the well known one.

It was interesting to note that this study could identify rhamnetin and isorhamnetin (difference in the position of the methoxy moiety) by the neutral and radical loss observed. The energy breakdown graphs were helpful to distinguish between these three pairs of structural isomers (including kaempferol and fisetin, and chyrsin and daidzein). All this could identify specific fragmentation pathways as future database information which will be helpful for quicker flavonoid product identification and their structural isomers.

Additionally, the competition of loss of methyl radical and methane were reported for rhamnetin and isorhamnetin in the negative mode.

Overall, the paper provides immense new information on fragmentation patterns for common flavonoids which is a good addition to the existing mass spectrometry literature.

Author Response

Comment 1: It would be interesting to mention the three diagnostic product ions identified for structural elucidation of unknown flavonoids and flavonoid metabolites within the abstract.

This has been added - shown in red text.

Comment 2: The reference style mentioned in the text doesn’t match the one in the reference section.

Something happened during the processing of the paper as the original was correctly numbered. This has now been addressed and 12 new references also added.

Comment 3: I would suggest to refer to a study for differentiation of isomers with methoxy substituents in ortho and para positions of the B-ring which was achieved using MS/MS in chalcones and flavanones.

We have concentrated on natural flavanoids in our study rather than synthetic ones, Although these two papers are interesting, in our view they are not directly relevant. If the editor believes we should add them then we can do that easily enough.

Comment 4: The suggested nomenclature is fine, as it is clearly demonstrated in the figures. But it would be good to understand why the authors decided to use this new one rather than the well known one.

We believe our nomenclature adds more differentiation than that presently in the literature.

Reviewer 2 Report

Comments and Suggestions for Authors

After reviewing the manuscripts titled "Systematic Characterisation of the Fragmentation of Flavonoids using High-Resolution Accurate Mass Electrospray Tandem Mass Spectrometry", I have the following comments.

1. The paper tackles an important topic in the analysis of flavonoids using advanced mass spectrometry (MS) techniques. The study helps towards understanding the structural elucidation of flavonoids, which are of great interest due to their biological activities.

2. While the authors provide a broad overview of the literature on flavonoid analysis using MS techniques, some references are rather outdated or generalized. For instance, the citations to methane chemical ionization (CI) studies from the 1970s (lines 45-47) could be supplemented with more recent comparisons or trends to give readers a more up-to-date perspective on advancements in the field

3.The literature review mainly draws on older studies, with a noticeable lack of very recent references (post-2020). Given the rapid developments in mass spectrometry, especially high-resolution techniques, integrating more recent studies would strengthen the scientific validity and relevance of the introduction

4. The results heavily rely on energy breakdown graphs (EBGs) to differentiate isomers. However, these graphs are presented with minimal explanation for non-specialist readers. The figures could benefit from an additional section explaining how to interpret the graphs.

5. The conclusion mentions potential applications in food safety and drug discovery, but there is little discussion throughout the paper that directly ties the results to these fields. Expanding on the practical applications or providing examples in these industries would make this claim more convincing .

Author Response

Comment 1: While the authors provide a broad overview of the literature on flavonoid analysis using MS techniques, some references are rather outdated or generalize.

Comment 2: .The literature review mainly draws on older studies

We have expanded the introduction with a broader discussion of more recent literature. In the process we have added 12 more references and we believe that this is now more up-to-date. This expanded text is shown in red.

Comment 3: The results heavily rely on energy breakdown graphs (EBGs) to differentiate isomers

We have rewritten and expanded the discussion section on EBGs - shown in red text.

Comment 4: The conclusion mentions potential applications

We have rewritten this paragraph - hopefully it is better now. Shown in red text. If the editor thinks that this paragraph is an issue, then we are happy to remove it.

Round 2

Reviewer 2 Report

Comments and Suggestions for Authors

After reviewing the the revised manuscript, I have the following comments:

1. The revisions made to the manuscript show satisfactory improvements. However, few minor issues still remain.

2. The energy breakdown graphs, could benefit from more detailed labeling or captions that summarize the results directly in the images, rather than in the main text. This would make it easier for readers to interpret the graphs at a glance.

3. There are instances where references to figures or tables display as "Error! Reference Source Not Found." Ensuring correct referencing throughout the manuscript is essential to maintain professionalism and usability.

4. The use of terms like "diagnostic ions" versus "product ions" may confuse readers without a clear, consistent distinction.

Overall, this study is well-executed and provides valuable insights into flavonoid fragmentation, presenting a comprehensive analysis with clear practical implications.

Author Response

Comment 1. The revisions made to the manuscript show satisfactory improvements. However, few minor issues still remain.

We hope these are now all addressed.

Comment 2. The energy breakdown graphs, could benefit from more detailed labeling or captions..

We spent sometime considering how to achieve this, and we have now recaptioned the EBGs to include much more information about the source of the ions and have highlighted the key ions that behave differenly to others. We hope that the EBGs look a lot beter now?

Comment 3. There are instances where references to figures or tables display as "Error! Reference Source Not Found.

I can find no instance of these errors in the paper. I am not sure what the reviewer is seeing that I am not. Can the editor please check that there are no issues?

Comment 4.  The use of terms like "diagnostic ions" versus "product ions" may confuse readers without a clear, consistent distinction.

We have address this wording usage to try to make it more definitive when we are talking about the key diagnostic ions as opposed to other structually indicative ions. This is shown in red in the text.